# Iodine-Catalyzed Cascade Annulation of 4-Hydroxycoumarins with Aurones: Access to Spirocyclic Benzofuran–Furocoumarins

**DOI:** 10.3390/molecules29081701

**Published:** 2024-04-09

**Authors:** Xuequan Wang, Changhui Yang, Dan Yue, Mingde Xu, Suyue Duan, Xianfu Shen

**Affiliations:** 1School of Chemistry and Resources Engineering, Honghe University, Honghe 661100, China; 2College of Chemistry and Environmental Science, Qujing Normal University, Qujing 655011, China

**Keywords:** iodine-catalyzed cascade annulation, spiro-heterocycle, benzofuran, furocoumarin

## Abstract

An attractive approach for the preparation of spirocyclic benzofuran–furocoumarins has been developed through iodine-catalyzed cascade annulation of 4-hydroxycoumarins with aurones. The reaction involves Michael addition, iodination, and intramolecular nucleophilic substitution in a one-step process, and offers an efficient method for easy access to a series of valuable spirocyclic benzofuran–furocoumarins in good yields (up to 99%) with excellent stereoselectivity. Moreover, this unprecedented protocol provides several advantages, including readily available materials, an environmentally benign catalyst, a broad substrate scope, and a simple procedure.

## 1. Introduction

Spiro-heterocycles are very important structural motifs in various natural products and pharmaceutical molecules, with a broad range of biological and pharmacological activities [1,2,3,4]. Considering the biological importance of spiro-heterocycles, a sufficiently large number of methods for the preparation of spiro-heterocycles have been developed, including the multicomponent tandem reaction [5], ring-expansion method [6], N-heterocyclic carbene (NHC)-catalyzed tandem annulation [7,8], palladium-catalyzed [3+2] cycloaddition [9] or 1,3-dipolar cycloaddition [10]. Recently, more eco-friendly electrochemical strategies have been used for the preparation of spiro-heterocycles. For example, Zhu et al. [11] developed an electrochemical method for the highly diastereoselective synthesis of spirocyclic indolines with significant anti-tumor activity.

In recent years, combining two rings to generate novel spiro-heterocycles has become an important approach to drug design [12,13]. The design and synthesis of biologically active spiro-heterocycles have attracted the attention of many pharmacologists and chemists [14,15]. In particular, the spiro-heterocycles that contain benzofuran have become one of the most interesting classes of molecules due to their notable biological activities [16,17,18]. For example, griseofulvin **A** is one of the earliest spirocyclic drugs to exhibit antifungal activity [19], while spiro-benzofuran **B**, with isobenzofuranone and benzofuranone motifs, has been found to be a core chemical skeleton for antivirals against influenza viruses [20]. (−)-Spiro-ganodermaine **G**, which is isolated from the *Ganoderma* species, displayed suppressive activity on the expression of TGF-β1-induced fibronectin and α-SMA, with a potential role in treating renal fibrosis (Figure 1) [21]. Moreover, furocoumarins demonstrate a wide range of biological activities [22,23] as anticancer [24], antioxidant [25], antifungal [26] and antiproliferative [27]. Therefore, there is great interest in exploring novel spiro-heterocycles between spiro-benzofurans and furocoumarins. Recently, Cui et al. [28] reported a novel procedure for spirocyclic benzofuran–furocoumarin synthesis, by Lewis acid-catalyzed [3+2]-cyclization of iodonium ylides with azadienes, in moderate yields with excellent stereoselectivity (Figure 1a). Meanwhile, Yavari et al. [29] developed an electrochemical approach for accessibility to spirocyclic benzofuran–furocoumarin (Figure 1b).

Molecular iodine, as a very simple and efficient reagent, has drawn considerable attention from synthetic chemists, and has been widely used in various organic transformations [30,31]. It is widely available, inexpensive, nontoxic, eco-friendly, and moisture resistant, and employed as a catalyst, resulting in the formation of new C–C [32,33], C–N [34,35], C–O [36], and C–S [37,38] bonds. In addition, molecular iodine has been recognized as a powerful tool for constructing the pharmacologically important heterocyclic rings [39,40,41]. Currently, several approaches have also been disclosed for the synthesis of spiro-heterocycles through iodine-mediated cascade reactions [42,43,44]. 

Based on the above discussion and our interest in exploring novel synthetic strategies for the construction of spiro-heterocycles, we developed an iodine-catalyzed cascade reaction of 4-hydroxycoumarins with aurones to afford spirocyclic benzofuran–furocoumarins with high stereoselectivity. This transformation displayed favorable compatibility for the preparation of various spirocyclic benzofuran–furocoumarins using I_2_ as the efficient and green catalyst.

## 2. Result and Discussion

Initially, the reaction of 4-hydroxycoumarin **1a** and aurone **2a** was investigated to optimize the reaction conditions. The experimental results are summarized in Table 1. Fortunately, the desired spirocyclic benzofuran–furocoumarin **3a** obtained a 54% yield from the reaction with 20 mol % of I_2_ at 80 °C (Table 1, entry 1). The structure of **3a** was determined based on their NMR spectroscopic similarities compared with the observed results [28,29]. In order to improve the efficacy of the reaction, different temperatures were evaluated for the reaction, and it was found that the yield of **3a** could be improved to 58% at 100 °C (Table 1, entry 2). Increasing the amount of I_2_ did not yield better results (Table 1, entry 4). Subsequently, the effect of an additive was investigated, such as *L*-proline, TBAI, and TEBAC. It was found that TEBAC afforded the desired product **3a** in better yields (Table 1, entries 5–7). Furthermore, increasing the amount of TEBAC loading to 40 mol % gave a higher yield of **3a** at 68% (Table 1, entry 9). Notably, changing the ratio of **2a**/**1a** from 1:1.3 to 1:1.5 (Table 1, entries 10–12) improved the yield of **3a** to 82% within 16 h (Table 1, entry 12). Therefore, the optimized reaction conditions for **3a** were 20 mol % of iodine as the catalyst, and 40 mol % of TEBAC as the additive, in DMSO at 100 °C for 16 h (Table 1, entry 12). 

With the optimal reaction conditions in hand, the substrate scope of 4-hydroxycoumarins and aurones were explored under standard reaction conditions. And the results are depicted in Figure 2. The 4-hydroxycoumarins bearing electron withdrawing (F, Cl, Br, NO_2_) or electron donating groups (OMe, Me) on Ar^1^ reacted smoothly, providing good yields of the corresponding spirocyclic benzofuran–furocoumarins, **3ab**–**3ah** and **3bl**. The reaction of 4-hydroxy-2*H*-benzo[*h*]chromen-2-one also proceeded smoothly and afforded an 86% yield of the spiro products **3ae**. For the aurones, both the electron withdrawing (F, Cl, Br) and electron donating groups (OMe, OEt, Me) on Ar^2^ of the benzofuranone moieties reacted with 4-hydroxycoumarin efficiently to afford 64−96% yields of the corresponding spiro products **3ai**–**3ao**. However, when the substrate bore a NO_2_ group on Ar^2^, the yield of the desired product **3bk** decreased to 62%. Subsequently, the effects of the substituents on the phenyl ring of Ar^3^ were evaluated. As the results reveal, the electron donating and electron withdrawing substituents of the phenyl ring on Ar^3^ have a substantial impact on the efficacy of the reaction. Generally, an electron donating group displayed respectable suitability with good to excellent yields. The weak electron donating groups (CH_3_) gave an 80% yield of **3ap**. Notably, substrates bearing a strong electron donating group (OMe, *^t^*Bu) on Ar^3^ showed excellent reactivity. The -o-OCH_3_, -p-OCH_3_, and -p-*^t^*Bu groups produced yields of 93%, 88%, and 94% for **3ar**, **3as** and **3aq**, respectively, while the substrates bearing three substituent groups of OCH_3_ gave the best results **3at** with a 99% yield. Adversely, some electron withdrawing groups decreased in reaction efficiency to afford the desired spiro products. For the substrates bearing the strong electron withdrawing groups (-p-NO_2_, -o-F), no desired spiro products were detected due to the competitive elimination pathway affording the coupled products **4au** and **4ax** with yields of 78% and 77%, respectively. While the COOMe and CN groups were tolerated, they only gave diminished yields of **3az** (51%) and **3ba** (56%), and a slight amount of the coupled products **4az** and **4ba** were observed. Interestingly, electron withdrawing groups, such as -m-NO_2_, -m-F, -p-F, Cl, Br, I, -p-CF_3_ and Ph, restored the activity and achieved good to excellent yields. Encouragingly, the heterocyclic furan and thiophene groups of Ar^3^ are also compatible and effectively furnished **3bi** and **3bj** with yields of 69% and 84%. A naphthyl group of Ar^3^ was also tolerated, and the desired spiro product **3bh** was isolated with an 89% yield. 

Just as Table 2 has shown, this approach for the synthesis of spirocyclic benzofuran–furocoumarins offers two advantages in terms of broad substrate scope and higher yields compared to reported methods. However, it still suffers from two drawbacks, including long reaction time and high temperature.

To improve the efficiency of this transformation, greener energy sources, including microwave and ultrasonic irradiation, were tested. As the results show, microwave irradiation displays obvious influence, which can improve the yield of **3a** from 7% to 30% compared with traditional protocol. However, when ultrasonic irradiation with a frequency of 35 KHz was used in the reaction, a 9% yield of **3a** was obtained, with a large amount of still unreacted substrate (Table 3).

Afterward, control experiments were conducted to investigate the mechanism of the formation of spirocyclic benzofuran–furocoumarins. First, the reaction of **1a** and **2a** was carried out without I_2_ under the optimum reaction conditions, and product **3a** was not detected (Figure 3a). Subsequently, substrates **1a** and **2a** were subjected to TsOH in DMSO at 100 °C, and no desired product was detected (Figure 3b). Finally, replacement of the solvent with MeCN allowed for trace amounts of the desired product **3a** (Figure 3c). The above results indicate that the formation of **3** is based on the cooperative effect of I_2_ and DMSO. 

According to the experimental results above and the previous literature [45], a plausible pathway is proposed in Figure 4. First, 4-hydroxycoumarin **1a** and aurone **2a** undergo Michael addition in the presence of I_2_ to generate intermediate **5**, which captures the proton released from the 4-hydroxycoumarin, thus forming intermediate **6**. In this process, TEBAC, as an effective catalyst following the addition of 4-hydroxycoumarin, is used to improve the efficiency of the Michael addition reaction [46]. Subsequently, intermediate **6** automerizes into its enol-form—intermediate **7**. Further electrophilic substitution between intermediate **7** and I_2_ is probably activated by DMSO and affords intermediate **8** immediately. Thereafter, an intramolecular nucleophilic substitution of intermediate **8** gives the expected product—the spirocyclic benzofuran–furocoumarins **3a**. During the iodination of **7** to **8** and the cyclization of **8** into **3a**, HI is both generated and then oxidized back into I_2_ by DMSO to accomplish the catalytic cycle. However, when intermediate **8** bears strong electron withdrawing groups, such as R_1_ = F or R_2_ = CN, NO_2_, COOMe, the competitive elimination pathway will be promoted in order to generate product **4**.

## 3. Materials and Methods

Unless otherwise specified, the starting materials and reagents used in the reactions were commercially available and used without further purification. Aurones **2** was prepared by the published procedures [47,48]. ^1^H (400 MHz), ^13^C (100 MHz), DEPT 135 (100 MHz) and DEPT 90 (100 MHz). NMR spectra were recorded on a Bruker Avance 400 spectrometer in CDCl_3_ or DMSO-*d_6_*. HRMS were performed with an AB QSTAR Pulsar mass spectrometer. Melting points were tested on an XT-4A melting-point apparatus without correction. The reactions were monitored by thin-layer chromatography (TLC) using silica gel GF254. For column chromatography, silica gel (200–300 mesh) was also employed.

The ^1^H NMR and ^13^C NMR spectral of the products are given in Appendix A.

### General Procedure

A mixture of 4-hydroxycoumarins **1** (0.375 mmol), aurones **2** (0.25 mmol), TEBAC (40 mol %), and I_2_ (20 mol %) was stirred in DMSO (0.5 mL) at 100 °C for 16 h; thereafter, saturated Na_2_S_2_O_3_ solution (8 mL) was added to quench the reaction. The product was then extracted with CH_2_Cl_2_ (3 × 8 mL). The combined organic layers were dried over anhydrous Na_2_SO_4_ and concentrated under reduced pressure. The crude product was subjected to flash column chromatography on silica gel (petroleum ether/ethyl acetate/CH_2_Cl_2_ = 10:1:5) to give 51–99% yields of the pure products **3aa**–**3bl**.

3′-Phenyl-3*H*,3′*H*,4′*H*-spiro[benzofuran-2,2′-furo[3,2-*c*]chromene]-3,4′-dione (**3aa**). Yield 82%; White solid; Mp 243–244 °C; ^1^H NMR (400 MHz, DMSO-*d_6_*): *δ* 7.85 (d, *J* = 7.6 Hz, 1H), 7.81–7.75 (m, 3H), 7.59 (d, *J* = 8.8 Hz, 1H), 7.47–7.43 (m, 1H), 7.31–7.28 (m, 4H), 7.25–7.21 (m, 2H), 7.06 (m, 1H), 5.14 (s, 1H); ^13^C NMR (100 MHz, DMSO-*d_6_*): *δ* 194.13 (C), 170.13 (C), 164.76 (C), 157.99 (C), 155.19 (C), 141.21 (CH), 134.30 (CH), 132.61 (C), 129.45 (CH), 128.73 (CH), 128.47 (CH), 125.95 (CH), 125.27 (CH), 124.85 (CH), 123.31 (CH), 118.10, 117.39 (CH), 113.55 (CH), 111.38 (C), 111.14 (C), 104.50 (C), 50.88 (CH); HRMS (ESI-TOF): *m*/*z* calcd for C_24_H_14_O_5_Na [M+Na]^+^: 405.0739, found: 405.0742.

8′-Methyl-3′-phenyl-3*H*,3′*H*,4′*H*-spiro[benzofuran-2,2′-furo[3,2-*c*]chromene]-3,4′-dione (**3ab**). Yield 79%; White solid; Mp 263–264 °C; ^1^H NMR (400 MHz, CDCl_3_): *δ* 7.66 (d, *J* = 7.6 Hz, 1H), 7.50 (t, *J* = 8.0 Hz, 1H), 7.42 (s, 1H), 7.36 (dd, *J* = 8.4, 1.6 Hz, 1H), 7.26 (d, *J* = 8.4 Hz, 1H), 7.18–7.17 (m, 3H), 7.09–7.06 (m, 3H), 6.74 (d, *J* = 8.4 Hz, 1H), 5.02 (s, 1H), 2.33 (s, 3H); ^13^C NMR (100 MHz, CDCl_3_): *δ* 194.36 (C), 170.46 (C), 165.38 (C), 158.76 (C), 153.56 (C), 139.84 (CH), 134.36 (CH), 134.22 (C), 131.49 (C), 128.88 (CH), 128.44 (CH), 128.21 (CH), 125.34 (CH), 123.68 (CH), 122.62 (CH), 118.46 (C), 116.86 (CH), 113.10 (CH), 111.37 (C), 110.94 (C), 103.77 (C), 52.06 (CH), 20.86 (CH_3_); HRMS (ESI-TOF): *m*/*z* calcd for C_25_H_16_O_5_Na [M+Na]^+^: 419.0895, found: 419.0896.

7′-Methyl-3′-phenyl-3*H*,3′*H*,4′*H*-spiro[benzofuran-2,2′-furo[3,2-*c*]chromene]-3,4′-dione (**3ac**). Yield 87%; Yellow solid; Mp 277–278 °C; ^1^H NMR (400 MHz, CDCl_3_): *δ* 7.67 (dd, *J* = 8.0, 0.8 Hz, 1H), 7.53–7.48 (m, 2H), 7.20–7.17 (m, 4H), 7.10–7.06 (m, 4H), 6.74 (d, *J* = 8.4 Hz, 1H), 5.01 (s, 1H), 2.42 (s, 3H); ^13^C NMR (100 MHz, CDCl_3_): *δ* 194.42 (C), 170.45 (C), 165.61 (C), 158.79 (C), 155.54 (C), 144.79 (C), 139.83 (CH), 131.56 (C), 128.86 (CH), 128.43 (CH), 128.19 (CH), 125.58 (CH), 125.34 (CH), 123.65 (CH), 122.66 (CH), 118.51 (C), 117.25 (CH), 113.09 (CH), 110.96 (C), 109.18 (C), 102.79 (C), 52.06 (CH), 22.12 (CH_3_); HRMS (ESI-TOF): *m*/*z* calcd for C_25_H_16_O_5_Na [M+Na]^+^: 419.0895, found: 419.0897.

7′-Methoxy-3′-phenyl-3*H*,3′*H*,4′*H*-spiro[benzofuran-2,2′-furo[3,2-*c*]chromene]-3,4′-dione (**3ad**). Yield 81%; Yellow solid; Mp 285–286 °C; ^1^H NMR (400 MHz, CDCl_3_): *δ* 7.67 (d, *J* = 8.0 Hz, 1H), 7.51 (t, *J* = 8.4 Hz, 2H), 7.19–7.17 (m, 3H), 7.09–7.06 (m, 3H), 6.85 (d, *J* = 1.6 Hz, 1H), 6.84–6.81 (m, 1H), 6.74 (d, *J* = 8.4 Hz, 1H), 5.00 (s, 1H), 3.83 (s, 3H); ^13^C NMR (100 MHz, CDCl_3_): *δ* 194.45 (C), 170.45 (C), 165.78 (C), 164.01 (C), 158.92 (C), 157.43 (C), 139.82 (CH), 131.68 (C), 128.86 (CH), 128.41 (CH), 128.15 (CH), 125.33 (CH), 124.02 (CH), 123.63 (CH), 118.53 (C), 113.08 (CH), 112.90 (CH), 110.97 (C), 104.90 (C), 100.90 (CH), 100.72 (C), 55.91 (CH), 51.96 (CH); HRMS (ESI-TOF): *m*/*z* calcd for C_25_H_16_O_6_Na [M+Na]^+^: 435.0845, found: 435.0846.

1′-Phenyl-1′*H*,3*H*,11′*H*-spiro[benzofuran-2,2′-benzo[*h*]furo[3,2-*c*]chromene]-3,11′-dione (**3ae**). Yield 86%; Yellow solid; Mp 268–269 °C; ^1^H NMR (400 MHz, CDCl_3_): *δ* 8.54 (dd, *J* = 6.4, 3.2 Hz, 1H), 7.84–7.82 (m, 1H), 7.68 (d, *J* = 7.6 Hz, 1H), 7.66–7.56 (m, 4H), 7.54–7.49 (m, 1H), 7.22–7.18 (m, 3H), 7.13–7.07 (m, 3H), 6.76 (d, *J* = 8.0 Hz, 1H), 5.10 (s, 1H); ^13^C NMR (100 MHz, CDCl_3_): *δ* 194.41 (C), 170.49 (C), 166.45 (C), 158.52 (C), 153.43 (C), 139.87 (CH), 135.51 (C), 131.48 (C), 129.35 (CH), 128.90 (CH), 128.48 (CH), 128.24 (CH), 128.10 (C), 127.52 (CH), 125.37 (CH), 124.55 (CH), 123.69 (CH), 123.10 (CH), 122.99 (C), 118.53 (C), 118.09 (CH), 113.11 (CH), 111.08 (C), 106.92 (C), 103.14 (C), 52.06 (CH); HRMS (ESI-TOF): *m*/*z* calcd for C_28_H_16_O_5_Na [M+Na]^+^: 455.0895, found: 455.0893.

8′-Fluoro-3′-phenyl-3*H*,3′*H*,4′*H*-spiro[benzofuran-2,2′-furo[3,2-*c*]chromene]-3,4′-dione (**3af**). Yield 77%; White solid; Mp 245–246 °C; ^1^H NMR (400 MHz, CDCl_3_): *δ* 7.67–7.65 (m, 1H), 7.53–7.49 (m, 1H), 7.37–7.34 (m, 1H), 7.31–7.25 (m, 2H), 7.20–7.17 (m, 3H), 7.10–7.06 (m, 3H), 6.75 (d, *J* = 8.4 Hz, 1H), 5.03 (s, 1H); ^13^C NMR (100 MHz, CDCl_3_): *δ* 194.05 (C), 170.43 (C), 164.57 (d, *J* = 3.0 Hz, C), 158.54 (d, *J* = 244.0 Hz, C), 158.11 (C), 151.50 (d, *J* = 2.0 Hz, C), 139.96 (CH), 131.11 (C), 128.89 (CH), 128.52 (CH), 128.36 (CH), 125.42 (CH), 123.84 (CH), 120.92 (d, *J* = 25.0 Hz, CH), 118.91 (d, *J* = 9.0 Hz, CH), 118.33 (C), 113.12 (CH), 112.39 (d, *J* = 10.0 Hz, C), 110.90 (C), 108.71 (d, *J* = 26.0 Hz, CH), 104.91 (C), 52.01 (CH); HRMS (ESI-TOF): *m*/*z* calcd for C_24_H_13_FO_5_Na [M+Na]^+^: 423.0645, found: 423.0644.

8′-Chloro-3′-phenyl-3*H*,3′*H*,4′*H*-spiro[benzofuran-2,2′-furo[3,2-*c*]chromene]-3,4′-dione (**3ag**). Yield 86%; White solid; Mp 278–279 °C; ^1^H NMR (400 MHz, CDCl_3_): *δ* 7.68–7.66 (m, 1H), 7.61 (d, *J* = 2.4 Hz, 1H), 7.54–7.49 (m, 2H), 7.32 (d, *J* = 8.8 Hz, 1H), 7.21–7.18 (m, 3H), 7.11–7.06 (m, 3H), 6.76 (d, *J* = 8.4 Hz, 1H), 5.03 (s, 1H); ^13^C NMR (100 MHz, CDCl_3_): *δ* 194.01 (C), 170.43 (C), 164.23 (C), 157.93 (C), 153.64 (C), 139.97 (CH), 133.28 (CH), 131.06 (C), 129.88 (C), 128.89 (CH), 128.53 (CH), 128.38 (CH), 125.44 (CH), 123.86 (CH), 122.53 (CH), 118.61 (CH), 118.30 (C), 113.13 (CH), 112.75 (C), 110.90 (C), 104.92 (C), 51.93 (CH); HRMS (ESI-TOF): *m*/*z* calcd for C_24_H_13_ClO_5_Na [M+Na]^+^: 439.0349, found: 439.0352.

8′-Bromo-3′-phenyl-3*H*,3′*H*,4′*H*-spiro[benzofuran-2,2′-furo[3,2-*c*]chromene]-3,4′-dione (**3ah**). Yield 65%; Yellow solid; Mp 286–287 °C; ^1^H NMR (400 MHz, CDCl_3_): *δ* 7.77 (d, *J* = 2.4 Hz, 1H), 7.68–7.63 (m, 2H), 7.52 (t, *J* = 7.6 Hz, 1H), 7.26 (d, *J* = 8.8 Hz, 1H), 7.21–7.19 (m, 3H), 7.11–7.06 (m, 3H), 6.77 (d, *J* = 8.4 Hz, 1H), 5.03 (s, 1H); ^13^C NMR (100 MHz, CDCl_3_): *δ* 193.98 (C), 170.43 (C), 164.09 (C), 157.85 (C), 154.10 (C), 139.95 (CH), 136.06 (CH), 131.06 (C), 128.89 (CH), 128.53 (CH), 128.37 (CH), 125.55 (CH), 125.43 (CH), 123.85 (CH), 118.85 (CH), 118.31 (C), 117.08 (C), 113.22 (C), 113.12 (CH), 110.90 (C), 104.92 (C), 51.91 (CH); HRMS (ESI-TOF): *m*/*z* calcd for C_24_H_13_BrO_5_Na [M+Na]^+^: 482.9844, found: 482.9848.

5-Methyl-3′-phenyl-3*H*,3′*H*,4′*H*-spiro[benzofuran-2,2′-furo[3,2-*c*]chromene]-3,4′-dione (**3ai**). Yield 80%; White solid; Mp 289–290 °C; ^1^H NMR (400 MHz, CDCl_3_): *δ* 7.62 (dd, *J* = 7.6, 1.2 Hz, 1H), 7.58–7.54 (m, 1H), 7.44 (s, 1H), 7.37 (d, *J* = 8.4 Hz, 1H), 7.32 (dd, *J* = 8.4, 1.6 Hz, 1H), 7.25 (t, *J* = 7.6 Hz, 1H), 7.21–7.17 (m, 3H), 7.08–7.06 (m, 2H), 6.65 (d, *J* = 8.4 Hz, 1H), 5.01 (s, 1H), 2.28 (s, 3H); ^13^C NMR (100 MHz, CDCl_3_): *δ* 194.40 (C), 168.94 (C), 165.43 (C), 158.58 (C), 155.33 (C), 141.05 (CH), 133.57 (C), 133.23 (CH), 131.50 (C), 128.87 (CH), 128.44 (CH), 128.19 (CH), 124.77 (CH), 124.30 (CH), 123.05 (CH), 118.33 (C), 117.12 (CH), 112.69 (CH), 111.73 (C), 111.24 (C), 103.91 (C), 52.01 (CH), 20.69 (CH_3_); HRMS (ESI-TOF): *m*/*z* calcd for C_25_H_16_O_5_Na [M+Na]^+^: 419.0895, found: 419.0900.

5-Methoxy-3′-phenyl-3*H*,3′*H*,4′*H*-spiro[benzofuran-2,2′-furo[3,2-*c*]chromene]-3,4′-dione (**3aj**). Yield 87%; White solid; Mp 269–270 °C; ^1^H NMR (400 MHz, CDCl_3_): *δ* 7.63 (d, *J* = 8.0 Hz, 1H), 7.58–7.54 (m, 1H), 7.37 (d, *J* = 8.4 Hz, 1H), 7.26 (t, *J* = 7.6 Hz, 1H), 7.20–7.18 (m, 3H), 7.13–7.04 (m, 4H), 6.67 (d, *J* = 9.2 Hz, 1H), 5.01 (s, 1H), 3.73 (s, 3H); ^13^C NMR (100 MHz, CDCl_3_): *δ* 194.64 (C), 165.79 (C), 165.43 (C), 158.56 (C), 156.04 (C), 155.32 (C), 133.25 (CH), 131.46 (C), 129.42 (CH), 128.85 (CH), 128.44 (CH), 128.23 (CH), 124.31 (CH), 123.04 (CH), 118.46 (C), 117.13 (CH), 114.03 (CH), 111.71 (C), 111.67 (C), 105.20 (CH), 103.93 (C), 56.01 (CH3), 52.13 (CH); HRMS (ESI-TOF): *m*/*z* calcd for C_25_H_16_O_6_Na [M+Na]^+^: 435.0845, found: 435.0848.

7-Methoxy-3′-phenyl-3*H*,3′*H*,4′*H*-spiro[benzofuran-2,2′-furo[3,2-*c*]chromene]-3,4′-dione (**3ak**). Yield 96% White solid; Mp 257–258 °C; ^1^H NMR (400 MHz, CDCl_3_): *δ* 7.62 (dd, *J* = 8.0, 1.6 Hz, 1H), 7.57–7.53 (m, 1H), 7.36 (d, *J* = 8.4 Hz, 1H), 7.26–7.23 (m, 2H), 7.19–7.17 (m, 3H), 7.12–7.10 (m, 2H), 7.04–6.97 (m, 2H), 5.04 (s, 1H), 3.57 (s, 3H); ^13^C NMR (100 MHz, CDCl_3_): *δ* 194.45 (C), 165.57 (C), 160.31 (C), 158.53 (C), 155.30 (C), 145.80 (C), 133.27 (CH), 131.23 (C), 128.93 (CH), 128.47 (CH), 128.24 (CH), 124.33 (CH), 124.17 (CH), 123.05 (CH), 122.70 (CH), 119.85 (C), 117.09 (CH), 116.60 (CH), 111.70 (C), 111.08 (C), 103.78 (C), 56.95 (CH3), 52.53 (CH); HRMS (ESI-TOF): *m*/*z* calcd for C_25_H_16_O_6_Na [M+Na]^+^: 435.0845, found: 435.0847.

6-Ethoxy-3′-phenyl-3*H*,3′*H*,4′*H*-spiro[benzofuran-2,2′-furo[3,2-*c*]chromene]-3,4′-dione (**3al**). Yield 64%; White solid; Mp 246–247 °C; ^1^H NMR (400 MHz, CDCl_3_): *δ* 7.76–7.72 (m, 1H), 7.69–7.63 (m, 2H), 7.50–7.45 (m, 1H), 7.39–7.28 (m, 4H), 7.21–7.17 (m, 2H), 6.72–6.67 (m, 1H), 6.24 (d, *J* = 9.6 Hz, 1H), 5.14 (d, *J* = 11.6 Hz, 1H), 4.07–4.00 (m, 2H), 1.45–1.39 (m, 3H); ^13^C NMR (100 MHz, CDCl_3_): *δ* 191.43 (C), 173.07 (C), 169.19 (C), 165.49 (C), 158.62 (C), 155.35 (C), 133.21 (CH), 131.64 (C), 128.90 (CH), 128.45 (CH), 128.15 (CH), 126.57 (CH), 124.30 (CH), 123.07 (CH), 117.11 (CH), 113.32 (CH), 112.02 (C), 111.77 (C), 111.11 (C), 103.92 (C), 96.69 (CH), 64.78 (CH_2_), 51.92 (CH), 14.40 (CH_3_); HRMS (ESI-TOF): *m*/*z* calcd for C_26_H_18_O_6_Na [M+Na]^+^: 449.1001, found: 449.1005.

6-Fluoro-3′-phenyl-3*H*,3′*H*,4′*H*-spiro[benzofuran-2,2′-furo[3,2-*c*]chromene]-3,4′-dione (**3am**). Yield 75%; White solid; Mp 224–225 °C; ^1^H NMR (400 MHz, CDCl_3_): *δ* 7.62 (dd, *J* = 7.6, 1.2 Hz, 1H), 7.59–7.54 (m, 1H), 7.37 (d, *J* = 8.4 Hz, 1H), 7.32–7.30 (m, 1H), 7.28–7.26 (m, 1H), 7.25–7.23 (m, 1H), 7.21–7.17 (m, 3H), 7.08–7.05 (m, 2H), 6.73–6.70 (m, 1H), 5.02 (s, 1H); ^13^C NMR (100 MHz, CDCl_3_): δ 193.99 (d, *J* = 2.0 Hz, C), 166.57 (C), 165.30 (C), 158.51 (d, *J* = 244.0 Hz, C), 158.42 (C), 155.34 (C), 133.37 (CH), 131.15 (C), 128.85 (CH), 128.52 (CH), 128.37 (CH), 127.37 (d, *J* = 26.0 Hz, CH), 124.39 (CH), 122.99 (CH), 119.05 (d, *J* = 8.0 Hz), 117.16 (CH), 114.39 (d, *J* = 8.0 Hz, CH), 111.67 (d, *J* = 14.0 Hz, C), 110.59 (d, *J* = 24.0 Hz, CH), 103.85 (C), 52.32 (CH); HRMS (ESI-TOF): *m*/*z* calcd for C_24_H_13_FO_5_Na [M+Na]^+^: 423.0645, found: 423.0646.

6-Chloro-3′-phenyl-3*H*,3′*H*,4′*H*-spiro[benzofuran-2,2′-furo[3,2-*c*]chromene]-3,4′-dione (**3an**). Yield 77%; White solid; Mp 265–266 °C; ^1^H NMR (400 MHz, CDCl_3_): *δ* 7.62–7.55 (m, 3H), 7.38 (d, *J* = 8.4 Hz, 1H), 7.26 (t, *J* = 7.2 Hz, 1H), 7.22–7.19 (m, 3H), 7.08–7.05 (m, 3H), 6.78 (s, 1H), 5.03 (s, 1H); ^13^C NMR (100 MHz, CDCl_3_): *δ* 192.75 (C), 170.55 (C), 165.25 (C), 158.40 (C), 155.36 (C), 146.25 (C), 133.37 (CH), 131.05 (C), 128.85 (CH), 128.58 (CH), 128.44 (CH), 126.08 (CH), 124.74 (CH), 124.39 (CH), 122.97 (CH), 117.17 (CH), 117.03 (C), 113.73 (CH), 111.58 (C), 111.43 (C), 103.85 (C), 52.22 (CH); HRMS (ESI-TOF): *m*/*z* calcd for C_24_H_13_ClO_5_Na [M+Na]^+^: 439.0349, found: 439.0350.

5-Bromo-3′-phenyl-3*H*,3′*H*,4′*H*-spiro[benzofuran-2,2′-furo[3,2-*c*]chromene]-3,4′-dione (**3ao**). Yield 88%; White solid; Mp 250–251 °C; ^1^H NMR (400 MHz, CDCl_3_): *δ* 7.77 (d, *J* = 2.0 Hz, 1H), 7.62 (dd, *J* = 8.0, 1.6 Hz, 1H), 7.59–7.55 (m, 2H), 7.37 (d, *J* = 8.4 Hz, 1H), 7.26 (t, *J* = 7.6 Hz, 1H), 7.20–7.18 (m, 3H), 7.07–7.05 (m, 2H), 6.66 (d, *J* = 8.4 Hz, 1H), 5.02 (s, 1H); ^13^C NMR (100 MHz, CDCl_3_): *δ* 193.04 (C), 169.13 (C), 165.27 (C), 158.38 (C), 155.35 (C), 142.32 (CH), 133.39 (CH), 131.03 (C), 128.84 (CH), 128.56 (CH), 128.41 (CH), 127.79 (CH), 124.40 (CH), 122.98 (CH), 120.15 (C), 117.17 (CH), 116.28 (C), 114.86 (CH), 111.57 (C), 111.29 (C), 103.81 (C), 52.35 (CH); HRMS (ESI-TOF): *m*/*z* calcd for C_24_H_13_BrO_5_Na [M+Na]^+^: 482.9844, found: 482.9849.

3′-(*p*-Tolyl)-3*H*,3′*H*,4′*H*-spiro[benzofuran-2,2′-furo[3,2-*c*]chromene]-3,4′-dione (**3ap**). Yield 80%; White solid; Mp 232-233 °C; ^1^H NMR (400 MHz, CDCl_3_): *δ* 7.76 (dd, *J* = 7.6, 0.8 Hz, 1H), 7.72 (dd, *J* = 8.0, 1.6 Hz, 1H), 7.67–7.59 (m, 2H), 7.47 (d, *J* = 8.4 Hz, 1H), 7.37–7.33 (m, 1H), 7.18 (t, *J* = 7.6 Hz, 1H), 7.11–7.06 (m, 4H), 6.88 (d, *J* = 8.4 Hz, 1H), 5.10 (s, 1H), 2.31 (s, 3H); ^13^C NMR (100 MHz, CDCl_3_): *δ* 194.46 (C), 170.53 (C), 165.23 (C), 158.58 (C), 155.30 (C), 139.84 (CH), 137.92 (C), 133.21 (CH), 129.23 (CH), 128.77 (CH), 128.27 (C), 125.35 (CH), 124.30 (CH), 123.68 (CH), 123.01 (C), 118.44 (C), 117.12 (CH), 113.20 (CH), 111.73 (C), 110.93 (C), 104.15 (C), 51.69 (CH), 21.25 (CH_3_); HRMS (ESI-TOF): *m*/*z* calcd for C_25_H_16_O_5_Na [M+Na]^+^: 419.0895, found: 419.0899.

3′-(4-(tert-Butyl)phenyl)-3*H*,3′*H*,4′*H*-spiro[benzofuran-2,2′-furo[3,2-*c*]chromene]-3,4′-dione (**3aq**). Yield 94%; Yellow oil; ^1^H NMR (400 MHz, CDCl_3_): *δ* 7.66 (d, *J* = 8.0 Hz, 1H), 7.62 (dd, *J* = 7.6, 1.2 Hz, 1H), 7.57–7.48 (m, 2H), 7.36 (dd, *J* = 8.4, 2.8 Hz, 1H), 7.26–7.22 (m, 1H), 7.20–7.18 (m, 2H), 7.10–7.05 (m, 1H), 6.99 (d, *J* = 8.4 Hz, 2H), 6.75 (dd, *J* = 8.4, 3.2 Hz, 1H), 5.00 (s, 1H), 1.18 (s, 9H); ^13^C NMR (100 MHz, CDCl_3_): *δ* 194.48 (C), 170.55 (C), 165.23 (C), 158.61 (C), 155.29 (C), 150.93 (C), 139.75 (CH), 133.20 (CH), 128.46 (CH), 128.27 (C), 125.39 (CH), 125.34 (C), 124.30 (CH), 123.66 (CH), 123.02 (CH), 118.48 (C), 117.10 (CH), 113.16 (CH), 111.73 (C), 111.05 (C), 104.15 (C), 51.56 (CH), 34.54 (C), 31.29 (CH_3_); HRMS (ESI-TOF): *m*/*z* calcd for C_28_H_22_O_5_Na [M+Na]^+^: 461.1365, found: 461.1367.

3′-(2-Methoxyphenyl)-3*H*,3′*H*,4′*H*-spiro[benzofuran-2,2′-furo[3,2-*c*]chromene]-3,4′-dione (**3ar**). Yield 93%; Yellow solid; Mp 257–258 °C; ^1^H NMR (400 MHz, CDCl_3_): *δ* 7.74 (d, *J* = 7.6 Hz, 1H), 7.60 (d, *J* = 8.0 Hz, 1H), 7.58–7.49 (m, 2H), 7.39 (d, *J* = 8.4 Hz, 1H), 7.24 (t, *J* = 7.6 Hz, 1H), 7.20–7.18 (m, 1H), 7.15 (d, *J* = 4.8 Hz, 1H), 7.09 (d, *J* = 7.6 Hz, 1H), 6.90 (t, *J* = 7.2 Hz, 1H), 6.74 (d, *J* = 8.4 Hz, 1H), 6.57 (d, *J* = 8.0 Hz, 1H), 5.26 (s, 1H), 3.09 (s, 3H); ^13^C NMR (100 MHz, CDCl_3_): *δ* 194.36 (C), 170.26 (C), 165.70 (C), 159.12 (C), 156.51 (C), 155.33 (C), 138.90 (CH), 133.21 (CH), 129.13 (CH), 128.63 (CH), 125.02 (CH), 124.29 (CH), 123.36 (CH), 123.03 (CH), 121.02 (C), 120.62 (CH), 118.67 (C), 117.06 (CH), 112.76 (CH), 111.75 (C), 111.27 (C), 109.25 (CH), 102.31 (C), 53.93 (CH_3_), 46.66 (CH); HRMS (ESI-TOF): *m*/*z* calcd for C_25_H_16_O_6_Na [M+Na]^+^: 435.0845, found: 435.0848.

3′-(4-Methoxyphenyl)-3*H*,3′*H*,4′*H*-spiro[benzofuran-2,2′-furo[3,2-*c*]chromene]-3,4′-dione (**3as**). Yield 88%; Yellow solid; Mp 220–221 °C; ^1^H NMR (400 MHz, CDCl_3_): *δ* 7.65 (dd, *J* = 7.6, 0.8 Hz, 1H), 7.62 (dd, *J* = 8.0, 1.2 Hz, 1H), 7.57–7.49 (m, 2H), 7.36 (d, *J* = 8.4 Hz, 1H), 7.27–7.23 (m, 1H), 7.08 (t, *J* = 7.2 Hz, 1H), 7.03–6.99 (m, 2H), 6.78 (d, *J* = 8.4 Hz, 1H), 6.72 (d, *J* = 8.8 Hz, 2H), 4.98 (s, 1H), 3.67 (s, 3H); ^13^C NMR (100 MHz, CDCl_3_): *δ* 194.50 (C), 170.47 (C), 165.19 (C), 159.38 (C), 158.59 (C), 155.29 (C), 139.87 (CH), 133.23 (CH), 130.05 (CH), 125.32 (CH), 124.32 (CH), 123.68 (CH), 123.32 (C), 123.01 (CH), 118.47 (C), 117.11 (CH), 113.90 (CH), 113.19 (CH), 111.72 (C), 110.89 (C), 104.15 (C), 55.21 (CH), 51.47 (CH); HRMS (ESI-TOF): *m*/*z* calcd for C_25_H_16_O_6_Na [M+Na]^+^: 435.0845, found: 435.0847.

3′-(3,4,5-Trimethoxyphenyl)-3*H*,3′*H*,4′*H*-spiro[benzofuran-2,2′-furo[3,2-*c*]chromene]-3,4′-dione (**3at**). Yield 99%; White solid; Mp 279–280 °C; ^1^H NMR (400 MHz, CDCl_3_): *δ* 7.68 (dd, *J* = 7.6, 0.4 Hz, 1H), 7.65 (d, *J* = 8.0 Hz, 1H), 7.61–7.53 (m, 2H), 7.40 (d, *J* = 8.0 Hz, 1H), 7.30–7.26 (m, 1H), 7.13–7.09 (m, 1H), 6.79 (d, *J* = 8.4 Hz, 1H), 6.24 (s, 2H), 4.99 (s, 1H), 3.72 (s, 3H), 3.63 (s, 6H); ^13^C NMR (100 MHz, CDCl_3_): *δ* 194.40 (C), 170.61 (C), 165.66 (C), 158.62 (C), 155.35 (C), 153.10 (C), 140.04 (CH), 137.69 (C), 133.42 (CH), 126.80 (C), 125.26 (CH), 124.42 (CH), 123.74 (CH), 123.10 (CH), 118.52 (C), 117.16 (CH), 113.24 (CH), 111.68 (C), 110.87 (C), 105.85 (CH), 103.39 (C), 60.78 (CH_3_), 56.10 (CH_3_), 52.55 (CH); HRMS (ESI-TOF): *m*/*z* calcd for C_27_H_20_O_8_Na [M+Na]^+^: 495.1056, found: 495.1060.

3′-(3-Nitrophenyl)-3*H*,3′*H*,4′*H*-spiro[benzofuran-2,2′-furo[3,2-*c*]chromene]-3,4′-dione (**3av**). Yield 82%; White solid; Mp 239–240 °C; ^1^H NMR (400 MHz, CDCl_3_): *δ* 8.17 (d, *J* = 8.0 Hz, 1H), 8.09 (s, 1H), 7.77 (d, *J* = 7.6 Hz, 1H), 7.74–7.67 (m, 2H), 7.63 (t, *J* = 8.0 Hz, 1H), 7.54 (d, *J* = 7.6 Hz, 1H), 7.49 (t, *J* = 7.2 Hz, 2H), 7.38 (t, *J* = 7.6 Hz, 1H), 7.21 (t, *J* = 7.2 Hz, 1H), 6.86 (d, *J* = 8.4 Hz, 1H), 5.20 (s, 1H); ^13^C NMR (100 MHz, CDCl_3_): *δ* 193.44 (C), 170.13 (C), 165.93 (C), 158.39 (C), 155.45 (C), 148.26 (C), 140.23 (CH), 135.28 (CH), 133.94 (C), 133.79 (CH), 129.53 (CH), 125.68 (CH), 124.61 (CH), 124.23 (CH), 124.09 (CH), 123.52 (CH), 123.20 (CH), 118.19 (C), 117.30 (CH), 113.14 (CH), 111.44 (C), 110.34 (C), 103.01 (C), 51.25 (CH); HRMS (ESI-TOF): *m*/*z* calcd for C_24_H_13_NO_7_Na [M+Na]^+^: 450.0590, found: 450.0588.

3′-(3-Fluorophenyl)-3*H*,3′*H*,4′*H*-spiro[benzofuran-2,2′-furo[3,2-*c*]chromene]-3,4′-dione (**3aw**). Yield 85%; White solid; Mp 206–207 °C; ^1^H NMR (400 MHz, CDCl_3_): *δ* 7.75 (d, *J* = 7.6 Hz, 1H), 7.70 (d, *J* = 7.6 Hz, 1H), 7.68–7.60 (m, 2H), 7.46 (d, *J* = 8.4 Hz, 1H), 7.34 (t, *J* = 7.6 Hz, 1H), 7.24–7.17 (m, 2H), 6.99–6.93 (m, 2H), 6.92–6.89 (m, 1H), 6.87 (d, *J* = 8.4 Hz, 1H), 5.09 (s, 1H); ^13^C NMR (100 MHz, CDCl_3_): δ 193.98 (C), 170.41 (C), 165.61 (C), 162.71 (d, *J* = 244.0 Hz), 158.45 (C), 155.37 (C), 140.05 (CH), 133.97 (d, *J* = 7.0 Hz), 133.50 (CH), 129.99 (d, *J* = 8.0 Hz, CH), 125.49 (CH), 124.71 (d, *J* = 3.0 Hz, CH), 124.45 (CH), 123.93 (CH), 123.10 (CH), 118.30 (C), 117.19 (CH), 116.01 (d, *J* = 52.0 Hz, CH), 115.41 (d, *J* = 20.0 Hz, CH), 113.15 (CH), 111.55 (C), 110.64 (C), 103.45 (C), 51.51 (d, *J* = 2.0 Hz, CH); HRMS (ESI-TOF): *m*/*z* calcd for C_24_H_13_FO_5_Na [M+Na]^+^: 423.0645, found: 423.0644.

3′-(4-Fluorophenyl)-3*H*,3′*H*,4′*H*-spiro[benzofuran-2,2′-furo[3,2-*c*]chromene]-3,4′-dione (**3ay**). Yield 97%; White solid; Mp 206–207 °C; ^1^H NMR (400 MHz, CDCl_3_): *δ* 7.75 (d, *J* = 7.2 Hz, 1H), 7.71 (d, *J* = 7.6 Hz, 1H), 7.67–7.60 (m, 2H), 7.46 (d, *J* = 8.4 Hz, 1H), 7.34 (t, *J* = 7.6 Hz, 1H), 7.20–7.13 (m, 3H), 6.96 (t, *J* = 8.4 Hz, 2H), 6.86 (d, *J* = 8.4 Hz, 1H), 5.09 (s, 1H); ^13^C NMR (100 MHz, CDCl_3_): *δ* 194.13 (C), 170.34 (C), 165.46 (C), 162.57 (d, *J* = 246 Hz, C), 158.53 (C), 155.34 (C), 140.04 (CH), 133.43 (CH), 130.60 (d, *J* = 9.0 Hz, CH), 127.20 (d, *J* = 4.0 Hz, C), 125.42 (CH), 124.43 (CH), 123.86 (CH), 123.07 (CH), 118.40 (C), 117.17 (CH), 115.52 (d, *J* = 22.0 Hz, CH), 113.12 (CH), 111.61 (C), 110.68 (C), 103.69 (C), 51.36 (CH); HRMS (ESI-TOF): *m*/*z* calcd for C_24_H_13_FO_5_Na [M+Na]^+^: 423.0645, found: 423.0647.

Methyl 4-(3,4′-dioxo-3*H*,3′*H*,4′*H*-spiro[benzofuran-2,2′-furo[3,2-*c*]chromen]-3′-yl)benzoate (**3az**). Yield 51%; White solid; Mp 228–229 °C; ^1^H NMR (400 MHz, CDCl_3_): *δ* 7.87 (d, *J* = 8.4 Hz, 2H), 7.68 (dd, *J* = 8.0, 0.8 Hz, 1H), 7.63 (dd, *J* = 7.6, 0.8 Hz, 1H), 7.60–7.56 (m, 1H), 7.54–7.50 (m, 1H), 7.38 (d, *J* = 8.4 Hz, 1H), 7.29–7.25 (m, 1H), 7.17 (d, *J* = 8.4 Hz, 2H), 7.10 (t, *J* = 7.2 Hz, 1H), 6.74 (d, *J* = 8.4 Hz, 1H), 5.08 (s, 1H), 3.80 (s, 3H); ^13^C NMR (100 MHz, CDCl_3_): *δ* 193.87 (C), 170.29 (C), 166.74 (C), 165.69 (C), 158.45 (C), 155.39 (C), 140.08 (CH), 136.69 (C), 133.51 (CH), 130.06 (C), 129.76 (CH), 129.02 (CH), 125.46 (CH), 124.46 (CH), 123.93 (CH), 123.10 (CH), 118.30 (C), 117.20 (CH), 113.12 (CH), 111.56 (C), 110.67 (C), 103.37 (C), 52.20 (CH3), 51.80 (CH); HRMS (ESI-TOF): *m*/*z* calcd for C_26_H_16_O_7_Na [M+Na]^+^: 463.0794, found: 463.0792.

4-(3,4′-Dioxo-3*H*,3′*H*,4′*H*-spiro[benzofuran-2,2′-furo[3,2-*c*]chromen]-3′-yl)benzonitrile (**3ba**). Yield 56%; Yellow solid; Mp 217-218 °C; ^1^H NMR (400 MHz, CDCl_3_): *δ* 7.69 (d, *J* = 7.6 Hz, 1H), 7.65–7.60 (m, 2H), 7.59–7.55 (m, 1H), 7.52–7.50 (m, 2H), 7.39 (d, *J* = 8.4 Hz, 1H), 7.29 (t, *J* = 7.6 Hz, 1H), 7.23 (d, *J* = 8.0 Hz, 2H), 7.13 (t, *J* = 7.6 Hz, 1H), 6.78 (d, *J* = 8.4 Hz, 1H), 5.06 (s, 1H); ^13^C NMR (100 MHz, CDCl_3_): *δ* 193.48 (C), 170.17 (C), 165.94 (C), 158.41 (C), 155.42 (C), 140.27 (CH), 137.09 (C), 133.76 (CH), 132.31 (CH), 129.79 (CH), 125.60 (CH), 124.60 (CH), 124.17 (CH), 123.16 (CH), 118.54 (C), 118.21 (C), 117.27 (CH), 113.11 (CH), 112.25 (C), 111.43 (C), 110.45 (C), 102.86 (C), 51.68 (CH); HRMS (ESI-TOF): *m*/*z* calcd for C_25_H_13_NO_5_Na [M+Na]^+^: 430.0691, found: 430.0689.

3′-(3-Chlorophenyl)-3*H*,3′*H*,4′*H*-spiro[benzofuran-2,2′-furo[3,2-*c*]chromene]-3,4′-dione (**3bb**). Yield 88%; Yellow solid; Mp 241–242 °C; ^1^H NMR (400 MHz, CDCl_3_): *δ* 7.70–7.67 (m, 1H), 7.64–7.61 (m, 1H), 7.60–7.53 (m, 2H), 7.38 (dd, *J* = 8.0, 3.6 Hz, 1H), 7.29–7.25 (m, 1H), 7.19–7.10 (m, 4H), 6.98 (d, *J* = 7.2 Hz, 1H), 6.81 (dd, *J* = 8.0, 3.2 Hz, 1H), 4.98 (s, 1H); ^13^C NMR (100 MHz, CDCl_3_): *δ* 193.92 (C), 170.37 (C), 165.57 (C), 158.40 (C), 155.37 (C), 140.04 (CH), 134.37 (C), 133.59 (C), 133.50 (CH), 129.71 (CH), 129.05 (CH), 128.60 (CH), 127.25 (CH), 125.52 (CH), 124.44 (CH), 123.95 (CH), 123.10 (CH), 118.29 (C), 117.21 (CH), 113.20 (CH), 111.55 (C), 110.62 (C), 103.47 (C), 51.42 (CH); HRMS (ESI-TOF): *m*/*z* calcd for C_24_H_13_ClO_5_Na [M+Na]^+^: 439.0349, found: 439.0350.

3′-(3-Bromophenyl)-3*H*,3′*H*,4′*H*-spiro[benzofuran-2,2′-furo[3,2-*c*]chromene]-3,4′-dione (**3bc**). Yield 72%; Yellow solid; Mp 260–261 °C; ^1^H NMR (400 MHz, CDCl_3_): *δ* 7.68 (dd, *J* = 8.0, 1.2 Hz, 1H), 7.64–7.60 (m, 1H), 7.58–7.53 (m, 2H), 7.38 (d, *J* = 8.4 Hz, 1H), 7.34–7.32 (m, 1H), 7.29–7.25 (m, 2H), 7.12 (t, *J* = 7.6 Hz, 1H), 7.08-7.02 (m, 2H), 6.81 (d, *J* = 8.4 Hz, 1H), 4.97 (s, 1H); ^13^C NMR (100 MHz, CDCl_3_): *δ* 193.91 (C), 170.36 (C), 165.56 (C), 158.39 (C), 155.37 (C), 140.05 (CH), 133.85 (C), 133.50 (CH), 131.91 (CH), 131.51 (CH), 129.99 (CH), 127.72 (CH), 125.52 (CH), 124.44 (CH), 123.96 (CH), 123.10 (CH), 122.56 (C), 118.29 (C), 117.21 (C), 113.21 (CH), 111.55 (C), 110.63 (C), 103.47 (C), 51.37 (CH); HRMS (ESI-TOF): *m*/*z* calcd for C_24_H_13_BrO_5_Na [M+Na]^+^: 482.9844, found: 482.9841.

3′-(4-Iodophenyl)-3*H*,3′*H*,4′*H*-spiro[benzofuran-2,2′-furo[3,2-*c*]chromene]-3,4′-dione (**3bd**). Yield 84%; White solid; Mp 255–256 °C; ^1^H NMR (400 MHz, CDCl_3_): *δ* 7.67 (dd, *J* = 7.6, 0.8 Hz, 1H), 7.62 (dd, *J* = 7.6, 0.8 Hz, 1H), 7.58 (dd, *J* = 7.6, 1.6 Hz, 1H), 7.55–7.51 (m, 3H), 7.38 (d, *J* = 8.4 Hz, 1H), 7.26 (t, *J* = 7.6 Hz, 1H), 7.11 (t, *J* = 7.6 Hz, 1H), 6.85 (d, *J* = 8.4 Hz, 2H), 6.81 (d, *J* = 8.4 Hz, 1H), 4.96 (s, 1H); ^13^C NMR (100 MHz, CDCl_3_): *δ* 193.97 (C), 170.36 (C), 165.58 (C), 158.46 (C), 155.35 (C), 140.09 (CH), 137.61 (CH), 133.47 (CH), 131.24 (C), 130.83 (CH), 125.45 (CH), 124.44 (CH), 123.93 (CH), 123.08 (CH), 118.30 (C), 117.19 (CH), 113.25 (CH), 111.57 (C), 110.56 (C), 103.44 (C), 94.22 (C), 51.50 (CH); HRMS (ESI-TOF): *m*/*z* calcd for C_24_H_13_IO_5_Na [M+Na]^+^: 530.9705, found: 530.9706.

3′-(2,4-Dichlorophenyl)-3*H*,3′*H*,4′*H*-spiro[benzofuran-2,2′-furo[3,2-*c*]chromene]-3,4′-dione (**3be**). Yield 64%; Yellow solid; Mp 253–254 °C; ^1^H NMR (400 MHz, CDCl_3_): *δ* 7.80 (d, *J* = 7.6 Hz, 1H), 7.72–7.65 (m, 3H), 7.49 (d, *J* = 8.4 Hz, 1H), 7.36 (t, *J* = 7.6 Hz, 1H), 7.32–7.30 (m, 1H), 7.28 (d, *J* = 2.4 Hz, 1H), 7.25 (t, *J* = 5.6 Hz, 2H), 6.93 (d, *J* = 8.4 Hz, 1H), 5.51 (s, 1H); ^13^C NMR (100 MHz, CDCl_3_): *δ* 193.70 (C), 170.39 (C), 165.90 (C), 158.47 (C), 155.40 (C), 139.69 (CH), 135.06 (C), 134.70 (C), 133.65 (CH), 130.88 (CH), 129.18 (CH), 129.08 (C), 127.35 (CH), 125.62 (CH), 124.52 (CH), 124.08 (CH), 123.15 (CH), 118.15 (C), 117.22 (CH), 113.08 (CH), 111.46 (C), 110.40 (C), 102.28 (C), 47.76 (CH); HRMS (ESI-TOF): *m*/*z* calcd for C_24_H_12_C_l2_O_5_Na [M+Na]^+^: 472.9959, found: 472.9958.

3′-(4-(Trifluoromethyl)phenyl)-3*H*,3′*H*,4′*H*-spiro[benzofuran-2,2′-furo[3,2-*c*]chromene]-3,4′-dione (**3bf**). Yield 90%; Yellow solid; Mp 220–221 °C; ^1^H NMR (400 MHz, CDCl_3_): *δ* 7.69 (dd, *J* = 7.6, 0.8 Hz, 1H), 7.64 (dd, *J* = 8.0, 1.6 Hz, 1H), 7.61–7.59 (m, 1H), 7.57–7.53 (m, 1H), 7.47 (d, *J* = 8.0 Hz, 2H), 7.39 (d, *J* = 8.4 Hz, 1H), 7.30–7.26 (m, 1H), 7.22 (d, *J* = 8.0 Hz, 2H), 7.12 (t, *J* = 7.6 Hz, 1H), 6.78 (d, *J* = 8.0 Hz, 1H), 5.08 (s, 1H); ^13^C NMR (100 MHz, CDCl_3_): *δ* 193.76 (C), 170.30 (C), 165.74 (C), 158.45 (C), 155.41 (C), 140.13 (CH), 135.65 (CH), 133.59 (C), 130.42 (q, *J* = 32.0 Hz, CF_3_), 129.35 (CH), 125.50 (q, *J* = 4.0 Hz, CH), 124.50 (CH), 124.02 (CH), 123.12 (CH), 118.26 (C), 117.23 (CH), 113.17 (CH), 111.53 (CH), 110.60 (CH), 103.25 (C), 51.52 (CH); HRMS (ESI-TOF): *m*/*z* calcd for C_25_H_13_F_3_O_5_Na [M+Na]^+^: 473.0613, found: 473.0616.

3′-([1,1′-Biphenyl]-4-yl)-3*H*,3′*H*,4′*H*-spiro[benzofuran-2,2′-furo[3,2-*c*]chromene]-3,4′-dione (**3bg**). Yield 88%; White solid; Mp 280–281 °C; ^1^H NMR (400 MHz, CDCl_3_): *δ* 7.68 (dd, *J* = 8.0, 0.8 Hz, 1H), 7.63 (dd, *J* = 7.6, 1.2 Hz, 1H), 7.58–7.54 (m, 1H), 7.52–7.50 (m, 1H), 7.49–7.47 (m, 1H), 7.46 (s, 1H), 7.43–7.41 (m, 2H), 7.38 (d, *J* = 8.0 Hz, 1H), 7.34–7.30 (m, 2H), 7.28–7.22 (m, 2H), 7.15 (d, *J* = 8.4 Hz, 2H), 7.10–7.06 (m, 1H), 6.77 (d, *J* = 8.4 Hz, 1H), 5.07 (s, 1H); ^13^C NMR (100 MHz, CDCl_3_): *δ* 194.32 (C), 170.50 (C), 165.44 (C), 158.61 (C), 155.36 (C), 140.97 (C), 140.55 (C), 139.92 (CH), 133.33 (CH), 130.47 (C), 129.31 (CH), 128.76 (CH), 127.40 (CH), 127.19 (CH), 127.06 (CH), 125.40 (CH), 124.38 (CH), 123.77 (CH), 123.07 (CH), 118.45 (C), 117.17 (CH), 113.23 (CH), 111.71 (C), 110.96 (C), 103.93 (C), 51.73 (CH); HRMS (ESI-TOF): *m*/*z* calcd for C_30_H_18_O_5_Na [M+Na]^+^: 481.1052, found: 481.1053.

3′-(Naphthalen-2-yl)-3*H*,3′*H*,4′*H*-spiro[benzofuran-2,2′-furo[3,2-*c*]chromene]-3,4′-dione (**3bh**). Yield 89%; White solid; Mp 257–258 °C; ^1^H NMR (400 MHz, CDCl_3_): *δ* 7.82–7.74 (m, 5H), 7.69 (s, 1H), 7.68–7.65 (m, 1H), 7.56–7.52 (m, 1H), 7.50–7.45 (m, 3H), 7.39–7.35 (m, 1H), 7.32 (dd, *J* = 8.4, 1.6 Hz, 1H), 7.15 (t, *J* = 7.6 Hz, 1H), 6.79 (d, *J* = 8.4 Hz, 1H), 5.32 (s, 1H); ^13^C NMR (100 MHz, CDCl_3_): *δ* 194.36 (C), 170.52 (C), 165.43 (C), 158.60 (C), 155.39 (C), 139.89 (CH), 133.35 (CH), 133.19 (C), 129.12 (C), 128.34 (CH), 128.21 (CH), 128.04 (CH), 127.67 (CH), 126.51 (CH), 126.21 (CH), 126.15 (CH), 125.40 (CH), 124.39 (CH), 123.77 (CH), 123.09 (CH), 118.31 (C), 117.18 (CH), 113.23 (CH), 111.73 (C), 111.03 (C), 104.07 (C), 52.05 (CH); HRMS (ESI-TOF): *m*/*z* calcd for C_28_H_16_O_5_Na [M+Na]^+^: 455.0895, found: 455.0895.

3′-(Furan-2-yl)-3*H*,3′*H*,4′*H*-spiro[benzofuran-2,2′-furo[3,2-*c*]chromene]-3,4′-dione (**3bi**). Yield 69%; Yellow solid; Mp 194–195 °C; ^1^H NMR (400 MHz, CDCl_3_): *δ* 7.70 (d, *J* = 7.6 Hz, 1H), 7.62–7.54 (m, 3H), 7.37 (d, *J* = 8.4 Hz, 1H), 7.27–7.23 (m, 2H), 7.14 (t, *J* = 7.6 Hz, 1H), 6.91 (d, *J* = 8.4 Hz, 1H), 6.26–6.23 (m, 2H), 5.11 (s, 1H); ^13^C NMR (100 MHz, CDCl_3_): *δ* 193.87 (C), 170.74 (C), 165.33 (C), 158.37 (C), 155.27 (C), 145.52 (C), 143.07 (CH), 139.94 (CH), 133.46 (CH), 125.55 (CH), 124.39 (CH), 123.92 (CH), 123.09 (CH), 118.12 (C), 117.14 (CH), 113.21 (CH), 111.56 (C), 110.61 (CH), 110.34 (C), 110.04 (CH), 101.82 (C), 45.50 (CH); HRMS (ESI-TOF): *m*/*z* calcd for C_22_H_12_O_6_Na [M+Na]^+^: 395.0532, found: 395.0537.

3′-(Thiophen-2-yl)-3*H*,3′*H*,4′*H*-spiro[benzofuran-2,2′-furo[3,2-*c*]chromene]-3,4′-dione (**3bj**). Yield 84%; Yellow solid; Mp 194–195 °C; ^1^H NMR (400 MHz, CDCl_3_): *δ* 7.68 (dd, *J* = 7.6, 0.8 Hz, 1H), 7.62 (dd, *J* = 8.0, 1.6 Hz, 1H), 7.60–7.55 (m, 2H), 7.37 (d, *J* = 8.4 Hz, 1H), 7.28–7.24 (m, 1H), 7.20–7.18 (m, 1H), 7.14–7.10 (m, 1H), 6.89 (d, *J* = 8.4 Hz, 1H), 6.86–6.84 (m, 2H), 5.33 (s, 1H); ^13^C NMR (100 MHz, CDCl_3_): *δ* 193.98 (C), 170.67 (C), 165.10 (C), 158.29 (C), 155.31 (C), 139.99 (CH), 133.68 (C), 133.47 (CH), 128.09 (CH), 126.72 (CH), 126.40 (CH), 125.46 (CH), 124.38 (CH), 123.87 (CH), 123.14 (CH), 118.39 (C), 117.17 (CH), 113.24 (CH), 111.57 (C), 110.14 (C), 103.89 (C), 47.15 (CH); HRMS (ESI-TOF): *m*/*z* calcd for C_22_H_12_O_5_SNa [M+Na]^+^: 411.0303, found: 411.0305.

5-nitro-3′-phenyl-3*H*,3′*H*,4′*H*-spiro[benzofuran-2,2′-furo[3,2-*c*]chromene]-3,4′-dione (**3bk**). Yield 62%; White solid; Mp 265–266 °C; ^1^H NMR (400 MHz, CDCl_3_): *δ* 8.57 (s, 1H), 8.41 (d, *J* = 9.2 Hz, 1H), 7.64-7.59 (m, 2H), 7.41 (d, *J* = 8.4 Hz, 1H), 7.30 (t, *J* = 7.6 Hz, 1H), 7.21-7.19 (m, 3H), 7.08 (s, 2H), 6.90 (d, *J* = 9.2 Hz, 1H), 5.10 (s, 1H); ^13^C NMR (100 MHz, CDCl_3_): *δ* 192.51 (C), 172.82 (C), 165.17 (C), 158.19 (C), 155.41 (C), 143.90 (C), 134.57 (CH), 133.62 (CH), 130.41 (C), 128.79 (CH), 128.72 (CH), 124.54 (CH), 122.90 (CH), 121.84 (CH), 119.01 (C), 117.27 (CH), 113.93 (CH), 112.15 (C), 111.38 (C), 103.66 (C), 52.86 (CH); HRMS (ESI-TOF): *m*/*z* calcd for C_24_H_13_NO_7_Na [M+Na]^+^: 450.0590, found: 450.0596.

8′-nitro-3′-phenyl-3*H*,3′*H*,4′*H*-spiro[benzofuran-2,2′-furo[3,2-*c*]chromene]-3,4′-dione (**3bl**). Yield 81%; Yellow solid; Mp 315–317 °C; ^1^H NMR (400 MHz, CDCl_3_): *δ* 8.56 (d, *J* = 2.4 Hz, 1H), 8.43 (dd, *J* = 9.2, 2.4 Hz, 1H), 7.70 (d, *J* = 7.6 Hz, 1H), 7.56 (t, *J* = 8.0 Hz, 1H), 7.51 (d, *J* = 9.2 Hz, 1H), 7.41-7.37 (m, 1H), 7.24-7.22 (m, 2H), 7.15-7.10 (m, 3H), 6.81 (d, *J* = 8.0 Hz, 1H), 5.06 (s, 1H); ^13^C NMR (100 MHz, CDCl_3_): *δ 1*93.65 (C), 170.44 (C), 164.07 (C), 158.40 (C), 156.86 (C), 143.77 (C), 140.12 (CH), 130.62 (C), 128.89 (CH), 128.64 (CH), 128.58 (CH), 127.80 (CH), 125.56 (CH), 124.07 (CH), 119.51 (CH), 118.35 (CH), 118.15 (C), 113.17 (CH), 112.06 (C), 110.92 (C), 105.86 (C), 51.78 (CH); HRMS (ESI-TOF): *m*/*z* calcd for C_24_H_13_NO_7_Na [M+Na]^+^: 450.0590, found: 450.0592.

4-Hydroxy-3-((4-nitrophenyl)(3-oxobenzofuran-2(3*H*)-ylidene)methyl)-2*H*-chromen-2-one (**4au**). Yield 78%; Yellow solid; Mp 238–239 °C; ^1^H NMR (400 MHz, CDCl_3_): *δ* 11.46 (s, 1H), 8.21 (d, *J* = 8.8 Hz, 2H), 7.96 (dd, *J* = 7.6, 1.2 Hz, 1H), 7.84 (dd, *J* = 8.0, 1.6 Hz, 1H), 7.67 (d, *J* = 8.8 Hz, 2H), 7.64–7.59 (m, 1H), 7.48–7.44 (m, 2H), 7.40 (t, *J* = 8.0 Hz, 1H), 6.98 (d, *J* = 8.4 Hz, 1H), 6.83–6.79 (m, 1H); ^13^C NMR (100 MHz, CDCl_3_): *δ* 186.27 (C), 163.66 (C), 158.81 (C), 156.53 (C), 153.71 (C), 148.10 (C), 147.67 (C), 137.47 (CH), 135.04 (C), 133.12 (CH), 131.76 (CH), 131.38 (CH), 131.03 (C), 125.26 (CH), 123.33 (CH), 121.86 (CH), 119.33 (CH), 118.83 (CH), 118.56 (C), 117.71 (CH), 111.66 (C), 110.01 (C); HRMS (ESI-TOF): *m*/*z* calcd for C_24_H_13_NO_7_Na [M+Na]^+^: 450.0590, found: 450.0589.

3-((2-Fluorophenyl)(3-oxobenzofuran-2(3*H*)-ylidene)methyl)-4-hydroxy-2*H*-chromen-2-one (**4ax**). Yield 77%; White solid; Mp 196–197 °C; ^1^H NMR (400 MHz, CDCl_3_): *δ* 7.80 (dd, *J* = 7.6, 0.8 Hz, 1H), 7.72–7.69 (m, 1H), 7.67–7.62 (m, 2H), 7.49 (d, *J* = 8.4 Hz, 1H), 7.38–7.34 (m, 1H), 7.32–7.26 (m, 2H), 7.24–7.20 (m, 1H), 7.16 (t, *J* = 7.2 Hz, 1H), 6.95 (t, *J* = 9.6 Hz, 1H), 6.89 (d, *J* = 8.4 Hz, 1H), 5.38 (s, 1H); ^13^C NMR (100 MHz, CDCl_3_): *δ* 193.80 (C), 170.39 (C), 158.67 (C), 155.34 (C), 139.67 (CH), 133.43 (CH), 130.01 (d, *J* = 8.0 Hz, CH), 125.58 (CH), 124.42 (CH), 124.25 (d, *J* = 3.0 Hz, CH), 123.90 (CH), 123.09 (CH), 119.38 (d, *J* = 14.0 Hz, (C)), 118.16 (C), 117.15 (CH), 115.09 (d, *J* = 22.0 Hz, CH), 113.00 (CH), 111.61 (C), 110.59 (C); HRMS (ESI-TOF): *m*/*z* calcd for C_24_H_13_FO_5_Na [M+Na]^+^: 423.0645, found: 423.0644.

Methyl-4-((4-hydroxy-2-oxo-2*H*-chromen-3-yl)(3-oxobenzofuran-2(3*H*)-ylidene)methyl)benzoate (**4az**). Yield 16%; Yellow solid; Mp 214–215 °C; ^1^H NMR (400 MHz, CDCl_3_): *δ* 11.51 (s, 1H), 8.02–8.00 (m, 2H), 7.96 (dd, *J* = 7.6, 1.2 Hz, 1H), 7.73 (dd, *J* = 8.0, 1.6 Hz, 1H), 7.62–7.58 (m, 1H), 7.56–7.53 (m, 2H), 7.45–7.41 (m, 2H), 7.40–7.36 (m, 1H), 6.96 (dd, *J* = 8.4, 0.8 Hz, 1H), 6.75–6.71 (m, 1H), 3.86 (s, 3H); ^13^C NMR (100 MHz, CDCl_3_): *δ* 186.79 (C), 166.60 (C), 163.44 (C), 158.75 (C), 156.62 (C), 153.67 (C), 147.32 (C), 137.22 (CH), 132.87 (CH), 132.84 (C), 132.07 (C), 131.95 (CH), 130.68 (C), 130.36 (CH), 129.42 (CH), 125.08 (CH), 121.86 (CH), 119.16 (CH), 118.61 (CH), 117.62 (CH), 111.82 (C), 110.09 (C), 52.32 (CH3), 29.74 (C); HRMS (ESI-TOF): *m*/*z* calcd for C_26_H_16_O_7_Na [M+Na]^+^: 463.0794, found: 463.0791.

4-((4-Hydroxy-2-oxo-2*H*-chromen-3-yl)(3-oxobenzofuran-2(3*H*)-ylidene)methyl)benzonitrile (**4ba**). Yield 26%; Yellow solid; Mp 259–260 °C; ^1^H NMR (400 MHz, CDCl_3_): *δ* 11.46 (s, 1H), 7.96 (d, *J* = 7.6 Hz, 1H), 7.78 (dd, *J* = 8.0, 0.8 Hz, 1H), 7.65–7.59 (m, 5H), 7.48–7.44 (m, 2H), 7.39 (t, *J* = 8.0 Hz, 1H), 6.99 (d, *J* = 8.8 Hz, 1H), 6.80–6.77 (m, 1H); ^13^C NMR (100 MHz, CDCl_3_): *δ* 186.41 (C), 163.59 (C), 158.82 (C), 156.55 (C), 153.70 (C), 147.53 (C), 137.41 (CH), 133.08 (C), 133.05 (CH), 131.86 (CH), 131.78 (CH), 131.28 (C), 131.08 (CH), 125.21 (CH), 121.85 (CH), 119.26 (CH), 118.78 (CH), 118.55 (C), 118.50 (C), 117.69 (CH), 113.01 (C), 111.69 (C), 109.93 (C); HRMS (ESI-TOF): *m*/*z* calcd for C_25_H_13_NO_5_Na [M+Na]^+^: 430.0691, found: 430.0691.

## 4. Conclusions

In summary, we developed a new approach to the synthesis of spirocyclic benzofuran–furocoumarins. The simple method utilizes readily available 4-hydroxycoumarins and aurones as materials and employs an iodine-catalyzed cascade annulation reaction to obtain a series of spirocyclic benzofuran–furocoumarins in high yields (up to 99%) with excellent stereoselectivity (up to >20:1 dr). Additionally, this operationally simple and environmentally benign strategy shows great compatibility with different groups on the 4-hydroxycoumarins and aurones. Further research on the application of this strategy in other reactions is underway in our laboratory.

## Data Availability

The data presented in this study are available in this article.

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
