# Peer review of "Iodine-Catalyzed Cascade Annulation of 4-Hydroxycoumarins with Aurones: Access to Spirocyclic Benzofuran–Furocoumarins"

_molecules, 2024, doi:10.3390/molecules29081701_

Round 1
Reviewer 1 Report
Comments and Suggestions for Authors
A correction has to be made concerning the proposed mechanism.
It is the double bond from the hydroxy coumarin which attacks the Beta position of the aurone. The aurone ends up as a 3-hydroxy benzofurane by capturing the proton released from the hydroxycoumarin changing to 4- ketocoumarin. The latter re-enolize to give compound 6. These first steps should be shown . It is helpful for less experienced people to understand the mechanism.
Concerning the last part, how to explain the entry of the iodine in position 2 of the benzofuran? Half of the iodine is missing then, as the HI formed is shown to come from the elimination needed for cyclisation .
Author Response
Thank you for your comments concerning our manuscript entitled “Iodine-Catalyzed Cascade Annulation of 4-Hydroxycoumarins with Aurones: Access to Spirocyclic Benzofuran-Furocoumarins” (No.: Molecules-2951171). Those comments are all valuable and very helpful for revising and improving our paper, as well as the important guiding significance to our researches. We have studied comments carefully and have made great changes which we hope meet with approval. Revised portion are marked in red in the manuscript.
|
Comments 1: A correction has to be made concerning the proposed mechanism.It is the double bond from the hydroxy coumarin which attacks the Beta position of the aurone. The aurone ends up as a 3-hydroxy benzofurane by capturing the proton released from the hydroxycoumarin changing to 4- ketocoumarin. The latter re-enolize to give compound 6. These first steps should be shown. It is helpful for less experienced people to understand the mechanism.
|
|
Response 1: We are very sorry that we do not make this clear in the manuscript. To make the reaction mechanism clearer, we have revised the manuscript. The possible intermediates and the regeneration of the I2 catalyst have been depicted clearly in Scheme 4. (Page 8, line 166)
|
|
Comments 2: How to explain the entry of the iodine in position 2 of the benzofuran? Half of the iodine is missing then, as the HI formed is shown to come from the elimination needed for cyclisation.
|
|
Response 2: I2/DMSO is a popular method for iodination on α position of ketone, and there are many reports about the application for the synthesis of heterocyclic compound. For our transformation, there are two positions which could be substituted by iodine. However, the position 3 of coumarin is too difficult to be replaced due to the special stability of conjugated structure. Moreover, HI is both generated during the iodination and the cyclization. However, I2 did not disappear for the regeneration of I2 catalyst from oxidation of HI by DMSO. Once HI is produced, it will be oxidized into I2 by DMSO immediately. |

Reviewer 2 Report
Comments and Suggestions for Authors
1. Overall, the manuscript has a lot of punctuation and grammatical errors and needs to be corrected. Besides, improve the language of the manuscript.
2. The abstract needs improvement.
3. Introduction section should be modified and a part related to the reported protocols for spirocyclic derivatives should added.
4. The authors recommended performing one representative example employing greener energy sources such as Microwave or Ultrasonic irradiation to illustrate the impacts on the reaction progression. Besides, make a comparison between the MW, US, and traditional protocols.
5. In addition, a comparative section (between the present protocol and report methodologies) is highly recommended.
Author Response
Thank you for your comments concerning our manuscript entitled “Iodine-Catalyzed Cascade Annulation of 4-Hydroxycoumarins with Aurones: Access to Spirocyclic Benzofuran-Furocoumarins” (No.: Molecules-2951171). Those comments are all valuable and very helpful for revising and improving our paper, as well as the important guiding significance to our researches. We have studied comments carefully and have made great changes which we hope meet with approval. Revised portion are marked in red in the manuscript.
|
Comments 1: Overall, the manuscript has a lot of punctuation and grammatical errors and needs to be corrected. Besides, improve the language of the manuscript.
|
|
Response 1: Special thanks for your comments and suggestions. The English language of our manuscript has been checked and revised repeatedly. We hope it could meet the journal's desired standard. In the case of future submissions, we will pay more attention to the English language and try my best to submit my papers written in standard, clear and grammatically correct English.
|
|
Comments 2: The abstract needs improvement.
|
|
Response 2: Thanks for the benefit advice from the reviewer. In the revised manuscript, we have revised the abstract as “An attractive approach for preparation of spirocyclic benzofuran-furocoumarins has been developed through iodine-catalyzed cascade annulation of 4-hydroxycoumarins with aurones. The reaction involves Michael addition, iodination and intramolecular nucleophilic substitution in one-step process, and offers an efficient method for easy access to a series of valuable spirocyclic benzofuran-furocoumarins in good yields (up to 99%) with excellet stereoselectivity. Moreover, this unprecedented protocol provides several advantages, including readily available materials, environmentally benign catalyst, broad substrate scope, and simple procedure.” |
|
|
|
Comments 3: Introduction section should be modified and a part related to the reported protocols for spirocyclic derivatives should added.
|
|
Response 3: Thanks for the benefit advice from the reviewer. According to the suggestion, the reported protocols for spirocyclic derivatives have added in the revised manuscript. In the revised manuscript, we have added the introduction as “Considering the biological importance of spiroheterocycles, a sufficiently large number of methods for the praparation of spiro-heterocycles have been developed including multicomponent tandem reaction [5], ring-expansion method [6], N-heterocycliccarbene (NHC) catalyzed tandem annulation [7, 8] , the plladium-catalyzed [3+2] cycloaddition [9] or 1,3-dipolar cycloaddition [10]. Recently, eco-friendly electrochemical strategy has been used for the preparation of spiroheterocycles. For example, Zhu and co-workers disclosed an electrochemical method for the highly diastereoselective synthesis of spirocyclic indolines with significant anti-tumor activity [11].” (Page 1, Line 23-31) Meanwhile, in the revised manuscript, the updated references related to the protocols for spirocyclic derivatives have been added in the Reference section. |
|
|
|
Comments 4: The authors recommended performing one representative example employing greener energy sources such as Microwave or Ultrasonic irradiation to illustrate the impacts on the reaction progression. Besides, make a comparison between the MW, US, and traditional protocols.
|
|
Response 4: Special thanks for your comments and suggestions. According to the suggestion, greener energy sources including microwave and ultrasonic irradiation have been tested for the transformation. And the comparison between MW, US and traditional protocols has been added in the revised manuscript. (Please see Table 3, Page 6.). |
|
Comments 5: In addition, a comparative section (between the present protocol and report methodologies) is highly recommended.
|
|
Response 5: Thanks for the constructive comments. The comparisons of the obtained results with literature data have been added in the revised manuscript. (Please see Table 2, Page 6.) |

Reviewer 3 Report
Comments and Suggestions for Authors
The manuscript "Iodine-Catalyzed Cascade Annulation of 4-Hydroxycoumarins with Aurones: Access to Spirocyclic Benzofuran- Furocoumarins" as an article in Molecules is well written and has the necessary quality to be considered in Molecules after some modifications:
1- In the reaction schemes, the reactants must be indicated above the reaction arrow, under the arrow to indicate the solvent, temperature and reaction time. This applies to scheme 1 and 2, and Table 1 scheme.
2- Correct typo: line 69, says "from 1:3 to 1:1.5". It should say "from 1:1.3 to 1:1.5".
3- Scheme 2 and 3 must be located in the text just after naming them, and not before being mentioned.
4- In the scope of substrates (Scheme 2), the authors analyzed the effect of electron-donating groups (Me, OMe, tBu, Ph, 2-furyl and thiophene) and electron-withdrawing groups (Cl, Br. I, F and CF3, CN, CO2Me NO2) in the Ar3 ring (according to the nomenclature used in scheme 2) and observed the formation of type 4 products. But, in the Ar1 and Ar2 rings only donor groups and weak electron-withdrawing groups were evaluated as (Cl, Br, I and F). Please include an example in Ar1 and Ar2 where a NO2 group is included, in order to evaluate the electronic effect of the other components of the reaction on the formation of product 4.
5- The proposed mechanism does not explain the need to use the TEBAC additive. Include a discussion of this effect supported by literature. As a suggestion, it would be a great advance in this work to present evidence of trapping of intermediates by ESI-MS monitoring.
6- Observing the 13C-NMR spectra used, I see that they used Dept 45 and Dept 90 spectra. Please identify the type of experiment used in section 3 ""materials and methods"
7- Since you have the information Dept 45 and Dept 90 in the supporting information, please in the 13C-NMR characterization of the compounds include in each signal, the type of crabono observed, that is, C, CH, CH2 or CH3.
Author Response
Thank you for your comments concerning our manuscript entitled “Iodine-Catalyzed Cascade Annulation of 4-Hydroxycoumarins with Aurones: Access to Spirocyclic Benzofuran-Furocoumarins” (No.: Molecules-2951171). Those comments are all valuable and very helpful for revising and improving our paper, as well as the important guiding significance to our researches. We have studied comments carefully and have made great changes which we hope meet with approval. Revised portion are marked in red in the manuscript.
|
Comments 1: In the reaction schemes, the reactants must be indicated above the reaction arrow, under the arrow to indicate the solvent, temperature and reaction time. This applies to scheme 1 and 2, and Table 1 scheme.
|
|
Response 1: Thanks for the benefit advice from the reviewer. According to the comments, we have revised the schemes in the manuscript.
|
|
Comments 2: Correct typo: line 69, says "from 1:3 to 1:1.5". It should say "from 1:1.3 to 1:1.5".
|
|
Response 2: Thanks a lot to the Reviewer! In the revised manuscript, the word “from 1:3 to 1:1.5” was revised as “from 1:1.3 to 1:1.5”. (Page 3, Line 82) |
|
|
|
Comments 3: Scheme 2 and 3 must be located in the text just after naming them, and not before being mentioned.
|
|
Response 3: In the revised manuscript, Scheme 2 and 3 have been located in the text just after naming them. |
|
|
|
Comments 4: In the scope of substrates (Scheme 2), the authors analyzed the effect of electron-donating groups (Me, OMe, tBu, Ph, 2-furyl and thiophene) and electron-withdrawing groups (Cl, Br. I, F and CF3, CN, CO2Me, NO2) in the Ar3 ring (according to the nomenclature used in scheme 2) and observed the formation of type 4 products. But, in the Ar1 and Ar2 rings only donor groups and weak electron-withdrawing groups were evaluated as (Cl, Br, I and F). Please include an example in Ar1 and Ar2 where a NO2 group is included, in order to evaluate the electronic effect of the other components of the reaction on the formation of product 4.
|
|
Response 4: Thanks for your kindly comments. According to the comment, substrates bearing a NO2 group on Ar1 or Ar2 rings were conducted, and the results were listed in Scheme 2. The results showed that Ar1 or Ar2 substituted with NO2 groups proceeded smoothly to afford the desired products with good yields. (Page 4-5, Scheme 2) |
|
Comments 5: The proposed mechanism does not explain the need to use the TEBAC additive. Include a discussion of this effect supported by literature. As a suggestion, it would be a great advance in this work to present evidence of trapping of intermediates by ESI-MS monitoring.
|
|
Response 5: Thanks for the constructive comments. According to the comment, the manuscript has been revised. And the explanation of the employment of TEBAC additive has been added in the text as “In this progress, TEBAC as an effective catalyst for the addition of 4-hydroxycoumarin was used to improve the efficiency of Michael addition [46].” The related reference has been added in the Reference section. (Page 7, Line 154-156) Moreover, trapping of intermediates is crucial to confirm the mechanism. However, that is a great challenge to us for lack of equipment conditions in our laboratory. Further research on the reaction mechanism is underway in our laboratory by synthesis of the possible intermediates. |
|
Comments 6: Observing the 13C-NMR spectra used, I see that they used Dept 45 and Dept 90 spectra. Please identify the type of experiment used in section 3 ""materials and methods"
|
|
Response 6: We are very sorry that we do not make this clear in the manuscript. In 13C-NMR spectra, Dept 135 and Dept 90 spectra were used to analyze the structure of products. We have presented the type of experiment in section 3 “materials and methods”. (Page 8, Line 171)
|
|
Comments 7: Since you have the information Dept 45 and Dept 90 in the supporting information, please in the 13C-NMR characterization of the compounds include in each signal, the type of crabono observed, that is, C, CH, CH2 or CH3. Response 7: Thanks for the benefit advice from the reviewer. In the revised manuscript, we have added the type of carbon to the data of 13C NMR. |

Reviewer 4 Report
Comments and Suggestions for Authors
In this study, the authors present a novel synthetic approach towards spirocyclic benzofuran-furocoumarins, which they used to prepare a library of new compounds with potent biological activity. However, several improvements are necessary before publication.
1) The authors mention the high stereoselectivity of the proposed methodology. Therefore, they should depict which diastereomer is formed predominantly and compare the observed results to those from citation 24.
2) The authors propose the mechanism of product formation; however, they do not mention how the side-products series 4 are formed.
3) The discussion comparing the proposed approach with the known ones (described in references 23 and 24), e.g., comparison of the yields, etc., is unfortunately missing.
4) The 13C NMR spectra for compound 3bf should be analyzed properly. In the list of signals, the authors omit the signal of CF3 and also do not present corresponding C-F coupling constants for the neighboring carbons.
Author Response
Thank you for your comments concerning our manuscript entitled “Iodine-Catalyzed Cascade Annulation of 4-Hydroxycoumarins with Aurones: Access to Spirocyclic Benzofuran-Furocoumarins” (No.: Molecules-2951171). Those comments are all valuable and very helpful for revising and improving our paper, as well as the important guiding significance to our researches. We have studied comments carefully and have made great changes which we hope meet with approval. Revised portion are marked in red in the manuscript.
|
Comments 1: The authors mention the high stereoselectivity of the proposed methodology. Therefore, they should depict which diastereomer is formed predominantly and compare the observed results to those from citation 24.
|
|
Response 1: Thanks for your kindly comments. We have compared with the observed results in the literature. According to their NMR spectroscopic similarities, the predominant diastereomer was determined, and their structures were depicted in the revised manuscript.
|
|
Comments 2: The authors propose the mechanism of product formation; however, they do not mention how the side-products series 4 are formed.
|
|
Response 2: Thanks for the kindly comments. To make the reaction mechanism clearer, we have revised the manuscript including the formation of side-products 4. (Page 8, Scheme 4) |
|
|
|
Comments 3: The discussion comparing the proposed approach with the known ones (described in references 23 and 24), e.g., comparison of the yields, etc., is unfortunately missing. |
|
Response 3: Thanks for the constructive comments. The comparisons of the obtained results with literature data have been added in the revised manuscript. (Please see Table 2, Page 6.) |
|
|
|
Comments 4: The 13C NMR spectra for compound 3bf should be analyzed properly. In the list of signals, the authors omit the signal of CF3 and also do not present corresponding C-F coupling constants for the neighboring carbons.
|
|
Response 4: Thanks for the benefit advice from the reviewer. The 13C NMR spectrum of 3bf has been reanalyzed. The signal of CF3 and C-F coupling constants for the neighboring carbons has been presented as “193.76 (C), 170.30 (C), 165.74 (C), 158.45 (C), 155.41 (C), 140.13 (CH), 135.65 (CH), 133.59 (C), 130.42 (q, J = 32.0 Hz, CF3), 129.35 (CH), 125.50 (q, J = 4.0 Hz, CH), 124.50 (CH), 124.02 (CH), 123.12 (CH), 118.26 (C), 117.23 (CH), 113.17 (CH), 111.53 (CH), 110.60 (CH), 103.25 (C), 51.52 (CH)” in the revised manuscript. (Page 13, Line 454-457) |
